# Background Summarization of Event Timelines

**Adithya Pratapa**[†]
Language Technology Institute
Carnegie Mellon University
vpratapa@cs.cmu.edu

**Kevin Small**      **Markus Dreyer**
Amazon
{smakevin,mddreyer}@amazon.com

## Abstract

Generating concise summaries of news events is a challenging natural language processing task. While journalists often curate timelines to highlight key sub-events, newcomers to a news event face challenges in catching up on its historical context. In this paper, we address this need by introducing the task of background news summarization, which complements each timeline update with a background summary of relevant preceding events. We construct a dataset by merging existing timeline datasets and asking human annotators to write a background summary for each timestep of each news event. We establish strong baseline performance using state-of-the-art summarization systems and propose a query-focused variant to generate background summaries. To evaluate background summary quality, we present a question-answering-based evaluation metric, Background Utility Score (BUS), which measures the percentage of questions about a current event timestep that a background summary answers. Our experiments show the effectiveness of instruction fine-tuned systems such as Flan-T5, in addition to strong zero-shot performance using GPT-3.5.[1]

## 1 Introduction

Events such as natural disasters, political conflicts, and elections are extensively covered by news agencies and followed by readers throughout the world. Generating concise summaries of these events is a challenging NLP task ([Chen et al., 2021](#)). For popular news stories, journalists curate retrospective timelines that highlight key sub-events on a timeline. However, for a new observer of a specific major event, catching up on the historical context needed to understand the significance of the sub-event update can be a challenging ordeal. To

this end, we present the task of background news summarization that complements each update in a timeline with a background summary.

Timeline summarization is the task of automatically extracting event timelines from a collection of news articles ([Chieu and Lee, 2004](#)). Update summarization ([Dang and Owczarzak, 2009](#)) involves summarizing a set of recent articles, assuming the reader is already familiar with a set of background articles. It has since been studied in shared tasks that track events in newswire ([Aslam et al., 2015](#)) and Twitter feeds ([Sequiera et al., 2018](#)). Our novel proposed task of background summarization presents an orthogonal use case for the update summarization task. A background summary provides *sufficient historical context* to the reader to help them understand the latest news update. It summarizes what has happened previously, in order to explain the background of the current news update.

Background summaries allow the reader to quickly grasp the historical context of an event without having to read through potentially hundreds of news articles or long timelines regarding a specific event. One application would be to contextualize short-text content (e.g., Tweets) with background information from news articles about the events. In addition to providing much-needed context, this can be useful for verifying the factuality of the events described in the tweet (e.g., Twitter Community Notes). In a news-centric conversational AI setting, a background summary may be generated to answer a user request to "tell me what I need to know to understand this event".

To construct a dataset for background summarization, we identify existing timeline datasets centered around major news events. Specifically, we select the widely-used Timeline17 ([Binh Tran et al., 2013](#)), Crisis ([Tran et al., 2015](#)), and Social Timeline ([Wang et al., 2015](#)) datasets – identifying 14 major news events from these datasets and prepare a single timeline of events for each major event.

---

[1]Code and data available at https://github.com/amazon-science/background-summaries

[†]work done during an internship at Amazon.

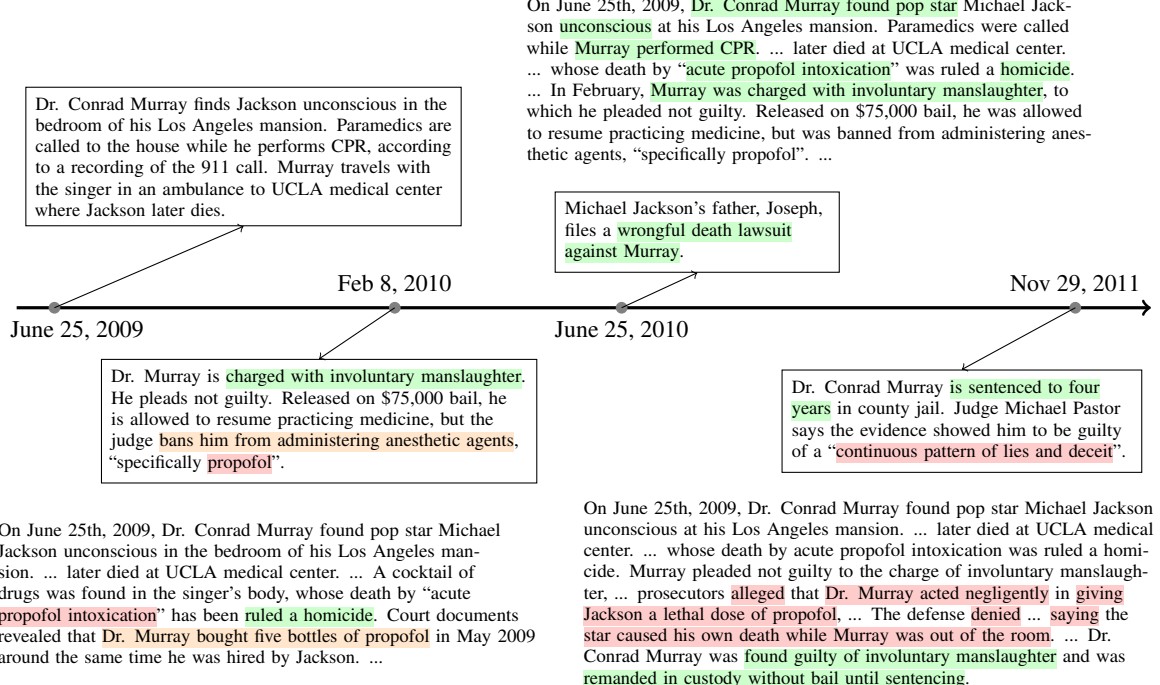

Figure 1: An illustration of the background summarization task. This is a snapshot from the timeline of the *Michael Jackson's Death* event. The timeline above shows four news updates between June 25, 2009, and November 29, 2011. Each update is complemented with a background summary that provides sufficient historical context to the events and entities described in the update. We highlight phrases from the background that provide context to specific phrases in the update text.

The original datasets included multiple timelines for major events, collected from different news agencies. For simplifying our annotation process, we merged all the timelines for a given major event to create a single timeline. We present these timelines to expert annotators and collect background summaries at each timestep for each news event. Figure 1 provides an example from the timeline of Michael Jackson's death. The timeline starts on June 25th, 2009 with a news update. The following updates on Feb 8, June 25, and Nov 29 are substantiated using background summaries that provide historical context regarding the event.

On the resulting annotated dataset, we experiment with a suite of summarization systems based on Flan-T5 (Chung et al., 2022), LongT5 (Guo et al., 2022), and GPT-3.5 (OpenAI, 2022). We propose to formulate the background summarization task as two different variants: (1) In a *generic* summarization setting, we generate a background summary for the current update at time step $t$ by using a concatenation of the past updates at time steps 1 through $t-1$. (2) In a *query-focused* setting (Dang, 2005; Xu and Lapata, 2022), we utilize the update at time step $t$ as a query. While the back-

ground summary does not include content from the current update, the current update can still be used for conditioning the summarization of past updates. This could potentially improve the utility of the backgrounds. For the query-focused summarization, we explore multiple variants of the query and present a comparison of Flan-T5 and GPT-3.5.

Our experiments indicate that the fine-tuned Flan-T5 system outperforms GPT-3.5 on the standard ROUGE metric while underperforming on factuality metrics. This illustrates the challenges associated with automatic metrics for summarization. Goyal et al. (2022) and Zhang et al. (2023) highlight the difficulties in using automatic metrics for comparing fine-tuned system summaries against zero-shot summaries from high-quality large language models (LLMs). Accordingly, we also present a novel question-answering-based evaluation of background summaries in this work that measures the utility of a given background summary to an update. First, we prompt GPT-3.5 to generate questions about the background of events and entities mentioned in the update. Second, we extract answers from the background summaries to measure their effectiveness in providing suffi-

cient historical context to the update. Our proposed Background Utility Score (BUS) measures the percentage of questions about current news updates that are answerable by a background summary. Our human evaluation results show the effectiveness of BUS. Our key contributions are as follows:

1. We present the new task of background summarization to help readers follow day-to-day updates on complex real-world events.

2. We describe an expert-annotated dataset covering 14 major news events from 2005–2014 with over 1,100 background summaries.

3. We conduct benchmark experiments using state-of-the-art zero-shot and fine-tuned summarization systems. We also explore query-focused summarization that generates the background by using the current update text as a query.

4. We propose an effective QA-based evaluation metric, Background Utility Score (BUS), to measure the utility of a background summary with respect to contextualizing an update.

## 2 Related Work

Events in the real world are often complex, consisting of numerous threads (Liu et al., 2017), and are reported by a large number of news agencies across the world. Tracking these events and providing important and useful updates to users has been the focus of many works in natural language processing and information retrieval communities (Chen et al., 2021). We highlight two specific variants of the event summarization task below,

**Timeline summarization:** Given a corpus of documents and a query, the task is to retrospectively extract important events from the documents and place them along a timeline (Chieu and Lee, 2004). A typical query consists of major events such as the Haitian earthquake or the BP oil spill. Datasets rely on timelines compiled by news journalists from agencies such as BBC, Reuters, and The New York Times, amongst others. Notable datasets for this task include Timeline17 (Binh Tran et al., 2013), Crisis (Tran et al., 2015), Social Timeline (Wang et al., 2015), entities dataset (Gholipour Ghalandari and Ifrim, 2020), and TLS-Newsroom (Born et al., 2020).

**Update summarization:** Dang and Owczarzak (2009) first proposed the task of update summarization. Given two sets of documents A & B, the task is to generate a query-focused update summary of the document set B assuming the user of the summary has already read the documents from set A. This task has since been studied on documents from newswire (Aslam et al., 2015) and Twitter feeds (Sequiera et al., 2018). In contrast to the timeline summarization task, systems do not have access to the documents from the future. Updating users about critical news events in real-time is very important to news and government agencies (Guo et al., 2013). However, comprehending these updates can be challenging for new readers. Our proposed task of background summarization serves a complimentary purpose to updates.

**Background summarization:** Hayashi et al. (2020) proposed the task of *disentangled paper summarization*, in which two separate summaries are generated for an academic paper: one summary describing the paper contribution, and another summarizing paper context. A related setting for news events can be a disentangled summarization of updates and backgrounds of events. Chen et al. (2022) presented a dataset of TV series transcripts and human-written recaps (SummScreen). Similar to a background in our task setup, recaps can help viewers understand the current episode. A key distinction is that a recap typically provides information from the most recent episode(s) only, but it does not provide general context to the story. In contrast, background summaries often include information from the very first update to put the current event in context.

Some prior works have studied the impact of background knowledge in the standard summarization task setup (Louis, 2014; Peyrard and West, 2020). A typical summarization setup requires a system to generate a summary of a collection of documents while ignoring any background knowledge already known to the receiver (or reader). Peyrard and West (2020) used a broader definition of background. In their setup, the background constitutes a document collection that the user is already familiar with, similar to the document set A from the above definition of update summarization. Our definition of background constitutes a summary of previous updates in a given event that are directly relevant to the current, most recent update.

| | Major event | Sources (# timelines) | Time period | # $U$ | $len(U)$ | $len(B)$ |
|---|---|---|---|---|---|---|
| train | Swine flu | T17 (3) | 2009 | 21 | 52 | 45 |
| | Financial crisis | T17 (1) | 2008 | 65 | 115 | 147 |
| | Iraq war | T17 (1) | 2005 | 155 | 41 | 162 |
| dev | Haitian earthquake | T17 (1) | 2010 | 11 | 100 | 61 |
| | Michael Jackson death | T17 (1) | 2009–2011 | 37 | 36 | 164 |
| | BP oil spill | T17 (5) | 2010–2012 | 118 | 56 | 219 |
| test | NSA leak | SocialTimeline (1) | 2014 | 29 | 45 | 50 |
| | Gaza conflict | SocialTimeline (1) | 2014 | 38 | 183 | 263 |
| | MH370 flight disappearance | SocialTimeline (1) | 2014 | 39 | 39 | 127 |
| | Yemen crisis | Crisis (6) | 2011–2012 | 81 | 30 | 125 |
| | Russian-Ukraine conflict | SocialTimeline (3) | 2014 | 86 | 112 | 236 |
| | Libyan crisis | T17 (2); Crisis (7) | 2011 | 118 | 38 | 177 |
| | Egyptian crisis | T17 (1); Crisis (4) | 2011–2013 | 129 | 34 | 187 |
| | Syrian crisis | T17 (4); Crisis (5) | 2011–2013 | 164 | 30 | 162 |

Table 1: An overview of the news events used in our background summarization dataset. The events are grouped into train, validation, and test splits. We list the source dataset and the number of source timelines for each event. The time period provides the overall span of the event timeline. The length of the timeline, the average word count of the (rewritten) updates, and newly annotated backgrounds are specified in the final columns.

**Long-form summarization:** Our proposed task often contains long timelines of events, requiring systems to perform long-form summarization. There is a growing effort in the community to improve long-range summarization systems. This includes works on book summarization (Wu et al., 2021), meeting summarization (Zhang et al., 2022), TV script summarization (Chen et al., 2022) and evaluation of long-form summarization systems (Krishna et al., 2023).

## 3 Background Summarization

Event timelines help readers keep track of updates regarding major news events. They provide a concise overview of the event's progress over time, without the need to read through hundreds or thousands of news articles written about the event. However, for long-lasting events, keeping track of all the sub-event threads can pose a major challenge for the user (Liu et al., 2017). We postulate that complementing each update with a short background summary regarding the event's past can assist the user in understanding the update. Our approach is inspired by the standard inverted pyramid structure of news articles (Pöttker, 2003). Typically, news articles consist of new newsworthy information at the top, followed by further details about the story, and end with necessary background information. This background information helps the reader gain

a full perspective of the news story. In this work, we extend this to news timelines.

### 3.1 Task

Given an event timeline consisting of a time series of updates $\langle U_1, \ldots, U_T \rangle$, the task is to generate background summaries $\langle B_2, \ldots, B_T \rangle$ for all updates after $U_1$. For each timestep $t > 1$, we wish to find the background summary $B_t$ that maximizes $p(B_t \mid U_1, \ldots, U_{t-1}; q)$ where $q$ is a query. In the generic baseline setting, $q$ is empty; in the query-focused setting, $q$ is set to the current update $U_t$. In the latter case, we do not aim to summarize $U_t$, but we use it to direct the summarization of the previous updates toward content that can help explain the current update $U_t$.

Note that each background summary $B_t$ is generated directly from the previous updates, independently from the previous background summary $B_{t-1}$. This enables us to include details relevant to $U_t$ from particular previous updates that may not be found in $B_{t-1}$.

### 3.2 Dataset Construction

To the best of our knowledge, there are no existing datasets that provide background summaries. Accordingly, we compile a new, expert-annotated dataset for this, building upon three popular news timeline summarization datasets, Timeline17

([Binh Tran et al., 2013](#)), Crisis ([Tran et al., 2015](#)), and Social Timeline ([Wang et al., 2015](#)).

**Timeline17:** compiled from an ensemble of news websites, this dataset provides 17 timelines spanning 9 major events from 2005–2013.

**Crisis:** a follow-up to the Timeline17 dataset, this covers 25 timelines spanning 4 major events. While it mostly covers a subset of events from Timeline17, it adds a new event (the Yemen crisis).

**Social Timeline:** compiled 6 timelines covering 4 major events from 2014. The timelines were collected from Wikipedia, NYTimes, and BBC.

Table 1 provides an overview of the 14 major news events compiled from the three datasets. Since the timelines were collected from various news websites (CNN, BBC, NYTimes, etc.), many events have more than one timeline. As each timeline covers the same underlying event, we merge them using timestamps to create a single timeline per event. During this merging process, we often end up with more than one update text per timestamp with possibly duplicate content. We ask the annotators to first rewrite the input updates to remove any duplicate content. Our annotation process for each news event contains the following three steps:

1. Read the input timeline to get a high-level understanding of the event.

2. For each timestep, read the provided 'rough' update summary. Rewrite the update into a short paragraph, removing any duplicate or previously reported subevents.

3. Go through the timeline in a sequential manner and write a background summary for each timestep.

Based on this process, we hired three professional annotators. For each timeline, we collect three independent (rewritten) update and (new) background pairs. Our full annotator guidelines are provided in Table 5 in the Appendix. Due to minor differences in the rewritten updates in the timelines, we do not merge the annotator timelines. Table 1 provides average lengths of rewritten updates and newly annotated background summaries for each major event. In our final dataset, each timestep in the timeline has three pairs of rewritten updates and background summaries.

|  | ROUGE-1 | ROUGE-2 | ROUGE-L |
|---|---|---|---|
| **Rewritten updates** | | | |
| Annotator 1 | 80.9 | 64.4 | 74.9 |
| Annotator 2 | 72.9 | 54.2 | 66.2 |
| Annotator 3 | 80.1 | 63.2 | 73.3 |
| **Background summaries** | | | |
| Annotator 1 | 47.9 | 21.3 | 43.3 |
| Annotator 2 | 44.9 | 16.6 | 39.5 |
| Annotator 3 | 48.0 | 21.1 | 43.4 |

Table 2: IAA across 14 major events.

### 3.3 Dataset Splits

For our experiments, we split the 14 major events into a train (3 events), validation (3 events), and test set (8 events). Table 1 lists the events in each split. We include a mixture of short and long timelines across the splits and the test set is mostly temporally separated from the train/dev splits.

Considering the strong few-shot summarization capabilities of large language models ([Goyal et al., 2022](#); [Zhang et al., 2023](#)), we decided to budget only a small fraction of expert-annotated data for training and development and leave most events to the test set. This allows sufficient data for further fine-tuning instruction-based models (Flan, GPT-3+) to our new task while maintaining sufficient diversity in the test set.

### 3.4 Inter-annotator Agreement

To measure the inter-annotator agreement (IAA), we compute ROUGE scores[2] with one annotator's summary as the hypothesis and the remaining two annotators' summaries as references.[3] Table 2 presents the IAA scores for both the re-written updates and the newly annotated backgrounds. As expected, we see high ROUGE scores on the rewritten updates. The scores are lower for background summaries, indicating the inherent variance in background summaries.

### 3.5 Background Utility Score (BUS)

Automatic metrics such as ROUGE are found to correlate poorly with human judgments of summaries ([Louis and Nenkova, 2013](#); [Peyrard, 2019](#)). Recent studies highlighted the ineffectiveness of standard metrics when comparing fine-tuned and

---

[2]For ROUGE-L, we use the Lsum variant in the Hugging-Face `evaluate` package.

[3]Multi-reference ROUGE returns a maximum score among references.

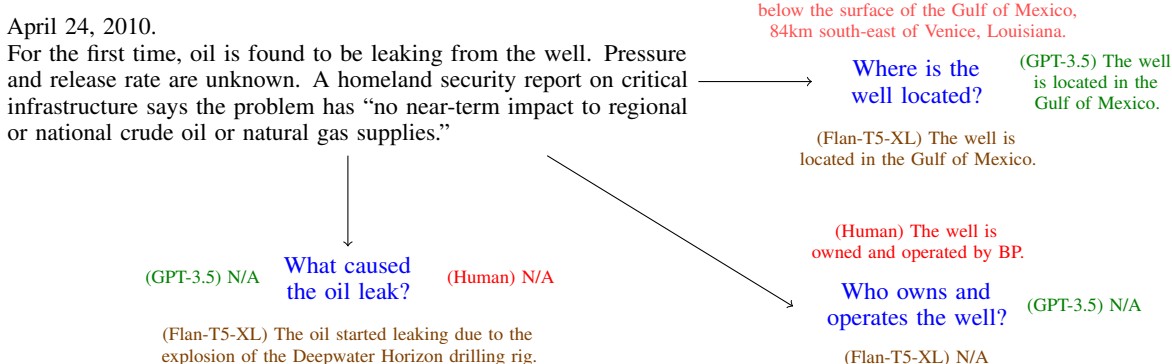

Figure 2: Examples of question-answer pairs for BUS (§3.5) generated by prompting GPT-3.5. This example shows an update text from the BP oil spill event. Questions are generated from the current update $U_t$, and the answers are generated based on three different background summaries $B_t$ (Human, GPT-3.5, Flan-T5-XL); N/A means the background summary did not provide an answer. The BUS score is calculated per system as the percentage of questions answered by its background summaries. See Table 8 in the Appendix for the full background summaries.

zero-shot summaries (Goyal et al., 2022; Zhang et al., 2023). To account for these limitations and the need to evaluate the quality of backgrounds, we propose a QA-based metric for the background summarization task. Our metric, Background Utility Score (BUS), measures the utility of a background $B_t$ to the corresponding update $U_t$.

To measure the utility, we first prompt a GPT-based model to generate (background) questions from the update text ($U_t$). We then re-prompt the model to extract answers from the background text ($B_t$). BUS measures the percentage of questions answerable by the background. Figure 2 presents examples of generated QA pairs. The background summary should be able to answer any questions the reader may have upon observing an update. While these questions are latent, we sample them by prompting a GPT-based model.

BUS is inspired by QuestEval (Scialom et al., 2021), an interpretable QA-based factuality metric for summarization. QuestEval measures the recall score by extracting questions from the source and computing the exact match $F_1$ between answer spans from the source and summary (vice-versa for precision). BUS is also tangentially related to recent LLM-based evaluation systems such as in Vicuna (Chiang et al., 2023) that explored the use of chatbots for evaluating chatbots.

## 4 Experiments

For our background summarization task, we experiment with three summarization systems, Flan-T5 (Chung et al., 2022), LongT5 (Guo et al., 2022),

and GPT-3.5 (OpenAI, 2022).

**Flan-T5:** an instruction fine-tuned version of T5 (Raffel et al., 2020). We use Flan-T5-XL with a maximum source length of 512 tokens.[4]

**LongT5:** a sparse attention variant of T5 that utilizes two efficient attentions, local and transient-global. Source length can be significantly longer than the standard 512 token limits of a T5-based system (§3.2). We use the Long-T5-TGlobal-XL with a maximum source length of 4096.[5]

**GPT-3.5:** a variant of the InstructGPT model (Ouyang et al., 2022) optimized for dialogue using reinforcement learning with human feedback. We use this model in a zero-shot setting. We set a maximum source length of 3696.[6]

We explore both generic and query-focused summarization settings (§3.1). In the query-focused setting, we use the current update ($U_t$) as an additional input to the summarization system.

**Generic:** We use a task prefix 'summarize:' to instruct T5-based systems. For GPT-3.5, we use a task suffix, 'Provide a short summary of the above article.'

**Query-focused:** The input for the T5-based systems follows the template, 'Generate a short query-focused summary of the background. Query:

---

[4]https://hf.co/google/flan-t5-xl
[5]https://hf.co/google/long-t5-tglobal-xl
[6]At the time of our experiments, this corresponds to the gpt-3.5-turbo-0301 version. https://platform.openai.com/docs/models/gpt-3-5

|  | ROUGE-1 | ROUGE-2 | ROUGE-L | QuestEval | BERTScore P | BUS–GPT-3.5 |
|---|---|---|---|---|---|---|
| **generic** | | | | | | |
| Flan-T5-XL | 43.5 \| 41.4 | 20.4 \| 17.4 | 39.9 \| 37.6 | 31.2 \| 25.0 | 86.3 \| 85.6 | 46.0 \| 42.2 |
| GPT-3.5 | 40.5 \| 37.7 | 15.5 \| 11.7 | 36.6 \| 33.0 | 37.2 \| 30.5 | 88.2 \| 87.2 | 59.1 \| 54.3 |
| **query-focused** | | | | | | |
| Flan-T5-XL | 43.0 \| 41.3 | 20.6 \| 17.4 | 39.5 \| 37.6 | 30.8 \| 24.9 | 86.2 \| 85.6 | 46.6 \| 43.6 |
| GPT-3.5 | 40.2 \| 40.5 | 15.4 \| 12.9 | 36.1 \| 35.9 | 36.9 \| 31.7 | 87.9 \| 87.5 | 49.9 \| 47.5 |

Table 3: System performance (dev | test) on the background summarization task.

<query>, Background: <past updates>.' For GPT-3.5, we use a task suffix, 'Generate a short query-focused summary of the background.' We use 512 and 128 limits for source and query. We consider two variants for queries. First, we use the full update ($U_t$) as the query. Second, we first extract named entities from and use those keywords as the query. The named entity-based approach removes any potential noise from the update and focuses solely on extracting background information about important persons or locations specified in the update. We use SpaCy English NER model to extract named entities from the query.

Across all our systems, when necessary we truncate the oldest updates from the input.[7] We train both Flan-T5 and LongT5 using DeepSpeed's ZeRO Stage 3 (Rasley et al., 2020). We set a maximum target length of 400 tokens.

**BUS:** We use GPT-3.5 as our question and answer generation system (ref. BUS–GPT-3.5). We generate five questions per update and use heuristic patterns on GPT answers to identify unanswerable questions.[8] Following recent work that showed better human alignment with GPT-4 (Liu et al., 2023), we also experiment with BUS–GPT-4.[9]

## 5 Results

### 5.1 Automatic Evaluation

Table 3 presents the results on validation and test sets for Flan-T5 and GPT-3.5 in both generic and the NER-based query-focused setups. We report scores on the standard summarization metric ROUGE (Lin, 2004), two factuality metrics QuestEval (Scialom et al., 2021), and BERTScore Precision (Zhang* et al., 2020; Pagnoni et al., 2021) and our proposed utility metric BUS (§3.5).

On the generic summarization setup, we observe that fine-tuned Flan-T5 outperforms zero-shot GPT-3.5 on ROUGE. However, the zero-shot GPT-3.5 model fares much better on factuality metrics and BUS. These trends are also valid in the NER-based query-focused formulation. Interestingly, we find the query-focused formulation generally underperforms the generic task.[10] In our experiments, we found Long-T5 underperforms Flan-T5 on the dev set (Table 9 in Appendix). We leave further evaluation of Long-T5-based systems for future work.

### 5.2 Human Evaluation

We conduct a human evaluation to determine the relative quality of the human-written backgrounds (subsection 3.2) and those generated by Flan-T5-XL and GPT-3.5 (generic; top-half of Table 3). We chose to evaluate the generic systems instead of query-focused systems due to their superior performance on ROUGE and factuality metrics on the development set.

**Setup:** We use the Amazon Mechanical Turk (AMT) platform. We sample 1,000 news updates from the test set and pair each one with the three background summaries, displayed in random order. We collect judgments from three annotators about which of the three displayed summaries is the best (i.e., most helpful) and which one is the worst (i.e., least helpful). We use majority voting to pick the best and worst summaries. Detailed instructions are shown Figure 5 in Appendix. Since annotators on the AMT platform are non-experts, we use multiple methods to obtain high-quality judgments, including a qualification test and time controls; details including fair compensation of the annotators are described in Appendix A.2.

**Results:** We use best-worst-scaling (BWS; Kiritchenko and Mohammad (2017)); Table 4 shows

---

[7] Other viable options are truncating middle updates or ranking updates based on their relevance to the current update.

[8] see Table 6 in Appendix for the instruction templates.

[9] We use gpt-3.5-turbo-0301 and gpt-4-0613.

[10] We present further analysis in A.5 in the Appendix.

| Human | Flan-T5-XL | GPT-3.5 |
|-------|------------|---------|
| 0.2430 | -0.0750 | -0.1680 |

Table 4: Results of the human evaluation on AMT using best-worst scaling (BWS). Values range from $-1$ (worst) to $+1$ (best).

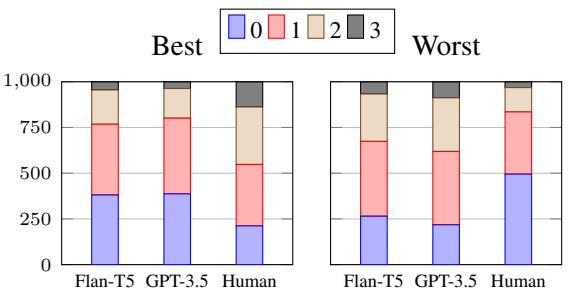

Figure 3: Vote distribution for best and worst systems from our human evaluation.

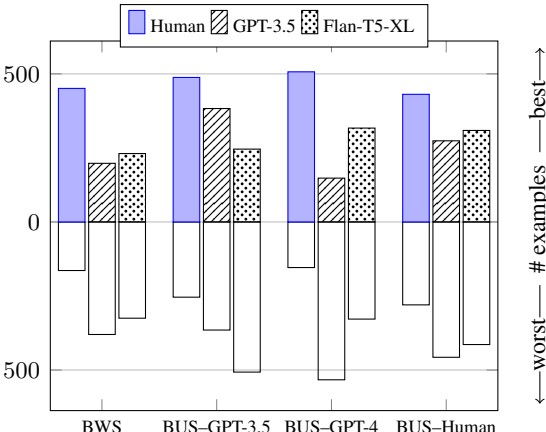

Figure 4: Aggregated best-worst votes for human-written, Flan-T5-XL, and GPT-3.5 backgrounds on the test set. The top and bottom halves report voted-best and voted-worst system counts respectively.

the results. The values are computed as the percentage of times a summary type is chosen as best minus the percentage of times it is selected as worst. Values of 1.0 or $-1.0$ indicate that the system has been unanimously picked as 'best' and 'worst' respectively. We observe that the human-written summaries are substantially preferred over both Flan-T5-XL and GPT-3.5 summaries.

**Agreement:** Figure 3 presents the vote distribution for the best and worst summaries across the 1k examples. Human-written backgrounds are rated the best by at least two annotators in 45% of the examples. They were rated the worst in less than 17% of the examples. Flan-T5 and GPT-3.5 have very similar best-vote distributions (23% and 20%). We see unanimous agreement on the best or worst system in less than 15% of the examples.

**Justifications:** Annotators tend to prefer human backgrounds over GPT-3.5's due to their comprehensiveness. In the justifications we collected as a part of our AMT evaluation, human backgrounds were described as 'most comprehensive', and providing 'complete context'. On the other hand, GPT-3.5 backgrounds were described as 'too short', 'just a timeline', and providing 'least information'.

# 6 BUS Analysis

Our human evaluation results showed variance amongst Turkers (Figure 3). This is in line with the observations made by prior work on standard summarization datasets (Goyal et al., 2022; Zhang et al., 2023). While human evaluation can be very useful,

past work highlighted the difficulties in choosing evaluation dimensions and task design (Khashabi et al., 2022). Goyal et al. (2022) recommends using an evaluation setup based on how users utilize the system in practice. To this end, we analyze the effectiveness of BUS (§3.5) in measuring the real-world utility of background summaries.

## 6.1 BUS–GPT

**Setup:** Following our human evaluation setup (§5.2), we compare human-written, Flan-T5-XL, and GPT-3.5 backgrounds. We compute the percentage of answerable questions using BUS (§3.5) and use this score to identify the best and worst systems for each update.

**Results:** Figure 4 provides the best-worst vote counts on the same 1,000 updates from test set using GPT-3.5-based BUS (ref. BUS–GPT-3.5) and GPT-4-based BUS (ref. BUS–GPT-4).[11] For comparison, we also include the vote counts from our human evaluation (§5.2; ref. BWS).[12]

With BUS–GPT-3.5, we observe that human-written backgrounds slightly outperform GPT-3.5. Flan-T5 significantly underperforms. BUS–GPT-4 is more closely aligned with our best-worst scaling human evaluation (BWS). This is in line with similar observations from prior work on GPT-4-based evaluation (Liu et al., 2023).

Overall, BUS–GPT-3.5 and BUS–GPT-4 exhibit different trends for human-written and GPT-

---

[11]For each example, we use BUS to designate one or more systems as best (or worst).

[12]See subsection A.5 in Appendix for event-level results.

3.5 backgrounds. To analyze this discrepancy, we present a BUS–human evaluation that uses question-and-answer pairs compiled by humans.

## 6.2 BUS–Human

Instead of relying on GPT-3.5 (or 4), we use Mechanical Turk to generate question-answer pairs. We first ask annotators to generate five background questions for each of the 1,000 news updates. For each of these tuples of update and questions, we pair it with one of the associated background summaries and ask annotators to attempt to answer all five questions using only information in one of the background summaries (or write *none* if the summary does not contain the answer). We then calculate BUS–Human as the percentage of answered questions per background summary type.[13] Results are presented in Figure 4 (ref. BUS–Human). BUS–Human shows clear alignment with our human evaluation results (BWS) and BUS–GPT-4, illustrating the effectiveness of our proposed BUS metric. However, this also highlights a potential drawback of using an automatic system such as GPT-3.5 for generating question-answer pairs.

## 6.3 Comparison of BUS methods

**Questions:** We analyze the questions generated by the three variants, BUS–GPT-3.5, BUS–GPT-4, and BUS–Human. In the Appendix, we provide questions generated for example updates from three test events, MH370 flight disappearance (Table 10), Yemen crisis (Table 11), and Libyan crisis (Table 12). Overall, both humans and GPT generate questions that specifically target background knowledge. Turkers' questions are specific and short, while GPT questions are more detailed and often contain two or more sub-questions. Questions target aspects such as named entities (Table 10) and past events (Table 11, Table 12).

However, we also see questions that do not target background information. Some questions from humans and GPT ask for additional details about events described in the update. See Q3 from Turker 2 and Q4 from GPT-3.5 in Yemen crisis (Table 11), and Q5 from GPT-4 in Libyan war (Table 12). A few questions ask about the consequences of the events described in the update. See Q5 from GPT-4 in MH370 disappearance (Table 10), Q4 & Q5 from Turker 2 in Yemen crisis (Table 11).

**BUS–GPT-3.5 vs BUS–GPT-4:** We notice BUS–GPT-3.5 suffers from answer hallucination, i.e., responds with an answer even if its not mentioned in the background text. On the other hand, GPT-4 is better at declining unanswerable questions (Table 13 in the Appendix). This explains our observation of better human alignment with BUS–GPT-4.

**BUS–Human vs BUS–GPT:** Our analysis indicates human evaluation remains the gold standard for our proposed background summarization task (BWS §5.2; BUS–Human §6.2). GPT-4 presents promising results and could serve as a fast, cost-effective alternative to human evaluation.

**Applications:** We believe BUS can be extended to related summarization tasks such as TV recaps (Chen et al., 2022) and disentangled summarization of scientific articles (Hayashi et al., 2020). A BUS-like metric can measure the relevancy of the recap to the current TV episode and the paper context to its contributions.

## 7 Conclusion & Future Work

To help readers follow long and complex event timelines, we propose the task of news background summarization. We compliment each update in the timeline with a background summary that provides sufficient context to the readers. We present an expert-annotated dataset for this task with over 1,100 background summaries from three annotators. On this dataset, we benchmark a suite of state-of-the-art summarization systems (Flan-T5, LongT5, and GPT-3.5). Our results show the zero-shot GPT-3.5 system outperforms the fine-tuned systems on the factuality metrics while underperforming on ROUGE. Given the lack of a metric that accurately captures the utility of a background summary to the news reader, we propose a novel QA-based metric, BUS, which measures the percentage of questions about the updates that are answerable from the respective background summaries.

For future work, we plan to explore background summarization directly from news articles instead of past updates. Sub-events previously considered unimportant but directly consequential to the latest news update can be captured in this setup. We are also interested in benchmarking aspect-based summarization systems for our task.

---

[13]Appendix A.3 contains more details about our setup, annotation guidelines, and compensation.

## Limitations

**Personalized Backgrounds:** While a background summary is helpful to any news reader, the utility can vary depending on the reader's familiarity with the event. In our BUS analysis (section 6), we observed differences in questions generated by two Turkers. In an ideal setting, systems should be capable of generating personalized background summaries catering to a reader.

**Local Events:** Our dataset and systems are currently limited to a selection of global, popular events involving disasters and conflicts (Table 1). For events local to a specific community, timelines are hard to find, and it's even harder to create background summaries. We acknowledge that backgrounds are equally impactful for local events and leave this extension to future work.

**Background from News Articles:** We generate backgrounds from past news updates. However, they can also be generated directly from news articles. We leave this extension to future work.

## Ethics Statement

We used Mechanical Turk for our human evaluation of summarization systems. We ensured prompt and fair pay to our annotators. We provided details about our selection criteria, per-task payment, and bonus in subsection A.2 and subsection A.3. Our background summarization dataset is expert-annotated on top of publicly available timeline summarization datasets. We protect our annotators' privacy and remove any personally identifiable information from our data release.

Like other text generation systems, generative summarization systems can suffer from hallucinations, potentially leading to misinformation. We acknowledge misinformation in backgrounds is undesirable in real-world applications. To this end, we report two factuality-based metrics to quantify the factuality of our systems.

## Acknowledgements

We thank the anonymous reviewers for their valuable feedback that helped improve our paper. We thank Teruko Mitamura for their support and Ting-Rui Chiang, Aidan San, and Xueguang Ma for helpful discussions in the early stages of this project.

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

# A   Appendix

## A.1   Annotation Guidelines for Writing Background Summaries

Table 5 presents the guidelines we presented to the annotators who wrote the summaries for our dataset (from subsection 3.2). We conducted multiple rounds of training with the annotators, where we reviewed annotator's work and provided feedback on the quality of background summaries.

## A.2   Details on the MTurk BWS Evaluation

We provide additional details on our Amazon Mechanical Turk setup (from subsection 5.2). We give detailed instructions to the annotators, see Figure 5. Workers who complete the tasks too quickly are automatically removed from our worker pool; their answers are replaced with new answers. We also use a bonus incentive structure. Every worker who passes the automatic time check receives a bonus at the end. In addition, we only use workers from our pool of about 300 trusted workers from previous studies. These were selected in two stages: (1) We only considered workers from countries whose main language is English and who have completed 100 or more HITs so far with an acceptance rate of 95% or higher. (2) In addition, workers must have passed an initial custom qualification test for a related text classification task we have conducted. Moreover, the resulting pool of workers has been used in more than 50 previous experiments, and we have over time removed any workers who have provided low-quality output in those previous experiments. On our batch of 1,000 HITs for the present human evaluation, we allowed any worker to complete a maximum of 333 HITs so that no worker can dominate the results. We used three annotators per HIT.

**Payment:**  We paid $0.70 per HIT with a bonus of $0.05 for all workers who passed automatic quality checks. 39 workers worked on our HITs overall, spending a median time of 169.0 seconds per HIT. This amounts to an average pay of $14.91 per hour per worker.

## A.3 Details on the MTurk BUS Evaluation

In order to calculate the BUS metrics based on human-written questions and answers (from subsection 6.2), we conducted two separate MTurk evaluations: (1) we obtained questions about news events and (2) we obtained answers to these questions given the different background summaries (human-written, or generated from GPT-3.5 or from Flan-T5). For both evaluations, we used the same general setup and annotator qualifications as described in §A.2.

To obtain five background questions for each of 1k news updates, we submitted 1,000 HITs. We paid $0.75 per HIT with a bonus of $0.05 for all workers who passed automatic quality checks. 46 workers worked on our HITs overall, spending a median time of 179.0 seconds per HIT. This amounts to an average pay of $15.08 per hour per worker. The annotation guidelines and an example annotation are shown in Figure 6. We allowed any worker to complete a maximum of 333 HITs so that no worker can dominate the results.

To obtain answers to the five questions per news update with respect to the three different background summaries, we submitted 3,000 HITs. We paid $0.70 per HIT with a bonus of $0.05 for all workers who passed automatic quality checks. 38 workers worked on our HITs overall, spending a median time of 144.2 seconds per HIT. This amounts to an average pay of $17.47 per hour per worker. The annotation guidelines and an example annotation are shown in Figure 7. We allowed any worker to complete a maximum of 500 HITs.

## A.4 Experiment Setup

**T5-based systems:** We perform training using DeepSpeed ZeRO stage 3 on two A6000 GPUs. We train the models for 10 epochs and pick the best model using the ROUGE-L score on the dev set. We use a per-device batch size of 8 and a learning rate of 1e-5. We use the Seq2SeqTrainer from Hugging Face in all of our experiments. At inference time, we use a beam size of 4, a length penalty of 2.0, and a no-repeat ngram size of 3.

**GPT-based systems:** We use the OpenAI python API for all of our GPT-based systems.

**Instructions for BUS–GPT:** Table 6 presents our instruction templates for question and answer generation using GPT models.

**Metrics:** ROUGE, BERTScore and QuestEval.[14,15,16]

## A.5 Additional Results

**Event-level BUS:** Similar to the results in Figure 4, we report the best-worst vote counts per event in the test set. For each event, we report counts for BWS (Figure 8), BUS–GPT-3.5 (Figure 9), BUS–GPT-4 (Figure 10) and BUS–Human (Figure 11).

**Query-focused Summarization:** In Table 3, our query-focused summarization setup did not provide gains. To analyze this behavior, we further experiment with an alternate query format where we use the full update text ($U_t$) as the query. Table 7 presents the results on the Flan-T5 system using ROUGE-L, QuestEval and BUS. We notice only a slight improvement in the performance when using full update text as the query.

---

[14]https://hf.co/spaces/evaluate-metric/rouge
[15]https://hf.co/spaces/evaluate-metric/bertscore
[16]https://github.com/ThomasScialom/QuestEval

Please read the above instructions carefully before you start the task. Below is an update from a news story.

**News Update:** Date: 2011-03-02, Article: The military briefly repels the rebels in Port Brega, an important oil-producing complex, but the rebels soon retake the city, with Gaddafi's forces retreating to Ras Lanuf.

For the above update, the following three summaries provide the *necessary historical context* from the news story. A good summary will provide relevant information from the past that helps you better understand the news update. Depending on their utility, identify the best (most helpful) and worst (least helpful) summaries.

**Background Summary 1:** Libyans went on strike on 30 January 2011 to protest the 40-year rule of Moammar Gaddafi, who had been in power since 1969. On 28 February, the EU banned the sale of arms and ammunition to Libya, and imposed a visa ban and freeze on the assets of Gadhafi and five of his family members. On 1 March, the UN General Assembly adopted a resolution to remove Libya from its seat on the 47-member Human Rights Council.

**Background Summary 2:** The article chronicles the events of the Libyan revolution in 2011, starting with peaceful demonstrations and escalating to violent clashes between protesters and security forces. The international community responds with condemnation, sanctions, and calls for Gaddafi to step down. Anti-Gaddafi rebels gain control of several cities, and the UN General Assembly removes Libya from its seat on the Human Rights Council.

**Background Summary 3:** Following the fall of Egyptian President Hisni Mubarak, protesters in Benghazi and other cities across Libya called for Libyan leader Moammar Gaddafi to step down. Gaddafi called in the army to suppress the rebellion, leading to much bloodshed which was condemned by the UN Security Council and the Arab League. As rebels took control of most eastern cities and closed in on Tripoli, international condemnation of the Libyan regime intensified, with the UN and EU imposing sanctions on the country, including an arms embargo.

## Rate the best and worst summaries.

| | | |
|---|---|---|
| **Best summary** | ☐ Summary 1  ☐ Summary 2  ☐ Summary 3 | |
| **Worst summary** | ☐ Summary 1  ☐ Summary 2  ☐ Summary 3 | |
| **Justification** | | |

Submit

Figure 5: This screenshot shows the human annotation interface to determine the best and the worst background summary for for best-worst scaling. In this example, the random order of displayed summaries is Flan-T5-XL, GPT-3.5, followed by the human-written summary. Here, both annotators marked the human-written summary as the best and the GPT-3.5 summary as the worst.

Please read the above instructions carefully before you start the task. Below is an update from a news story.

**News Update:** Date: 2011-05-01, Article: The NATO airstrike killed his youngest son, Seif al-Arab, and three grandchildren but it did not hurt Gaddafi.

Write five questions you might have about the background of this news story. Do not ask questions whose answer is contained directly in the above news update. We are interested in questions about the events from the past that are related to the news update.

## Background Questions

**Question 1**

What was the reason for the NATO airstrikes in Libya?

**Question 2**

Who was Seif al-Arab Gaddafi, and what was his role in the Libyan government?

**Question 3**

What were the circumstances of the airstrike that killed Seif al-Arab Gaddafi and his grandchildre

**Question 4**

What was the reaction of the Libyan government to the airstrike?

**Question 5**

What were the long-term consequences of the airstrike for the Libyan civil war?

Submit

Figure 6: This screenshot shows the annotation interface for MTurk annotators to write five questions about a news event. The questions from an annotator are shown in the text fields as an example.

Please read the above instructions carefully before you start the task. Below is a article from a news story.

**Article:** Demonstrators in Libya are demanding an end to Muammar Gaddafi's 42-year rule, taking part in large protests in Benghazi, the capital Tripoli, and several other cities around the country. These have been met with a strong-arm response from government security forces that has left hundreds dead and injured and earned international condemnation. With committed support, excluding troops on the ground, from the US, several European countries, the UN, and NATO, allied in pressing for the removal of Gaddafi's regime, rebels have been fighting government forces over control of strategic cities. In a speech, Gaddafi urged NATO to negotiate an end to airstrikes, accusing the coalition of killing civilians and destroying the nation's infrastructure in order to seize control of oil production. According to the government, a NATO missile attack on a house in Tripoli has killed Gaddafi's youngest son, Saif al-Arab, and three grandchildren.

For each question below, extract the answer from the above article. **If an answer is not available in the article, write "None".**

**Question 1**: What was the reason for the NATO airstrikes in Libya?

**Question 2**: Who was Seif al-Arab Gaddafi, and what was his role in the Libyan government?

**Question 3**: What were the circumstances of the airstrike that killed Seif al-Arab Gaddafi and his grandchildren?

**Question 4**: What was the reaction of the Libyan government to the airstrike?

**Question 5**: What were the long-term consequences of the airstrike for the Libyan civil war?

## Answers

**Answer 1**

> To remove the Gaddafi regime.

**Answer 2**

> He was Gaddafi's son, which his political role unspecified.

**Answer 3**

> It was a NATO missile attack on a house in Tripoli.

**Answer 4**

> Gaddafi urged NATO to negotiate to end the airstrikes.

**Answer 5**

> None

Submit

Figure 7: This screenshot shows the annotation interface for MTurk annotators to answer five questions about a news updates, given one of the background summaries. The answers from an annotator are shown in the text fields as an example.

**Terminology**

**Update:** a short text summary of *what's new* in the news story. This text summarizes the latest events, specifically ones that are important to the overall story.
**Background:** a short text summary that provides *sufficient historical context* for the current update. Background aims to provide the reader a quick history of the news story, without them having to read all the previous updates. Background should cover past events that help in understanding the current events described in the update.
**Timestep**: day of the event (YYYY-MM-DD).
**Timeline**: a series of timesteps. Each timestep in a timeline is associated with an update and a background summary.
**Super event**: the key news story or major event for which we are constructing a timeline. For instance, 'Egyptian Crisis', 'BP oil spill', 'MH 370 disappearance' are some of the super events from our dataset.

**Annotation Steps**

We follow a three-stage annotation process,
**Stage-0**: Read the input timeline to get a high-level understanding of the super-event.
**Stage-1**: For each timestep, read the provided 'rough' update summary. Rewrite the update into a short paragraph, removing any duplicate or previously reported subevents.
**Stage-2**: Go through the timeline in sequential manner and write background summaries for each timestep.

Table 5: Annotation guidelines for the background summarization task.

---

**BUS question generation**

Update: {update}
Imagine you read the above update about a news story. You have no prior information about the story. Generate five background questions you might have about the story.

**BUS answer extraction**

Background: {background}
Questions: {questions}
For each question, list answers from the background text when available. Say "unanswerable" if the question is not answered in the background text.

Table 6: Instruction templates for GPT-based question-answer generation.

| Flan-T5-XL | ROUGE-L | QuestEval | BUS |
|---|---|---|---|
| query: $U_t$ | 39.5 | 30.9 | 46.9 |
| query: NE($U_t$) | 39.5 | 30.8 | 46.6 |

Table 7: Ablation studies on different queries for background summarization task (dev set).

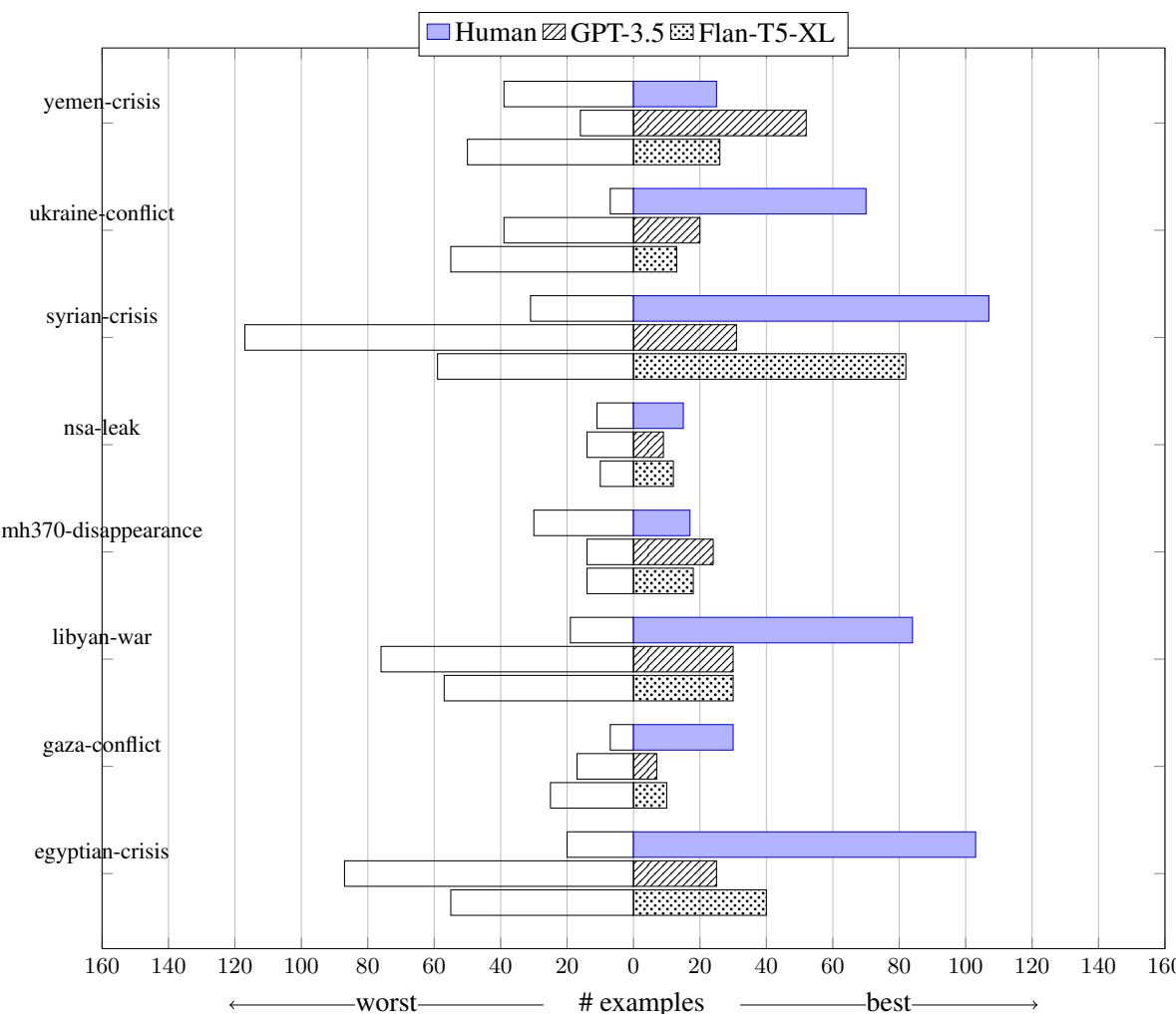

Figure 8: Aggregated best-worst votes for human-written, Flan-T5, GPT-3.5 backgrounds (BWS). The left and right halves report voted-worst and voted-best system counts respectively.

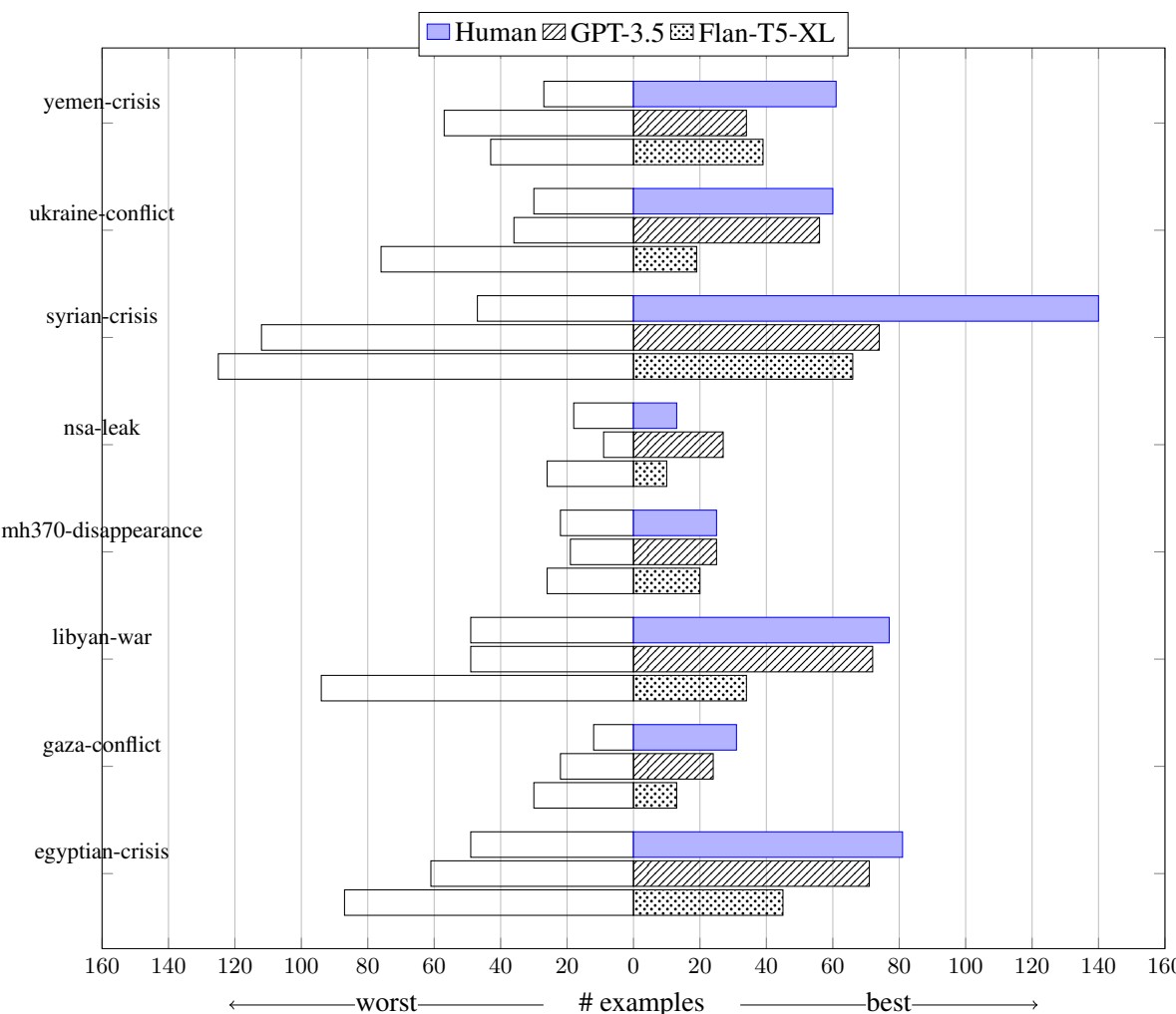

Figure 9: Aggregated best-worst votes for human-written, Flan-T5, GPT-3.5 backgrounds (BUS–GPT-3.5). The left and right halves report voted-worst and voted-best system counts respectively.

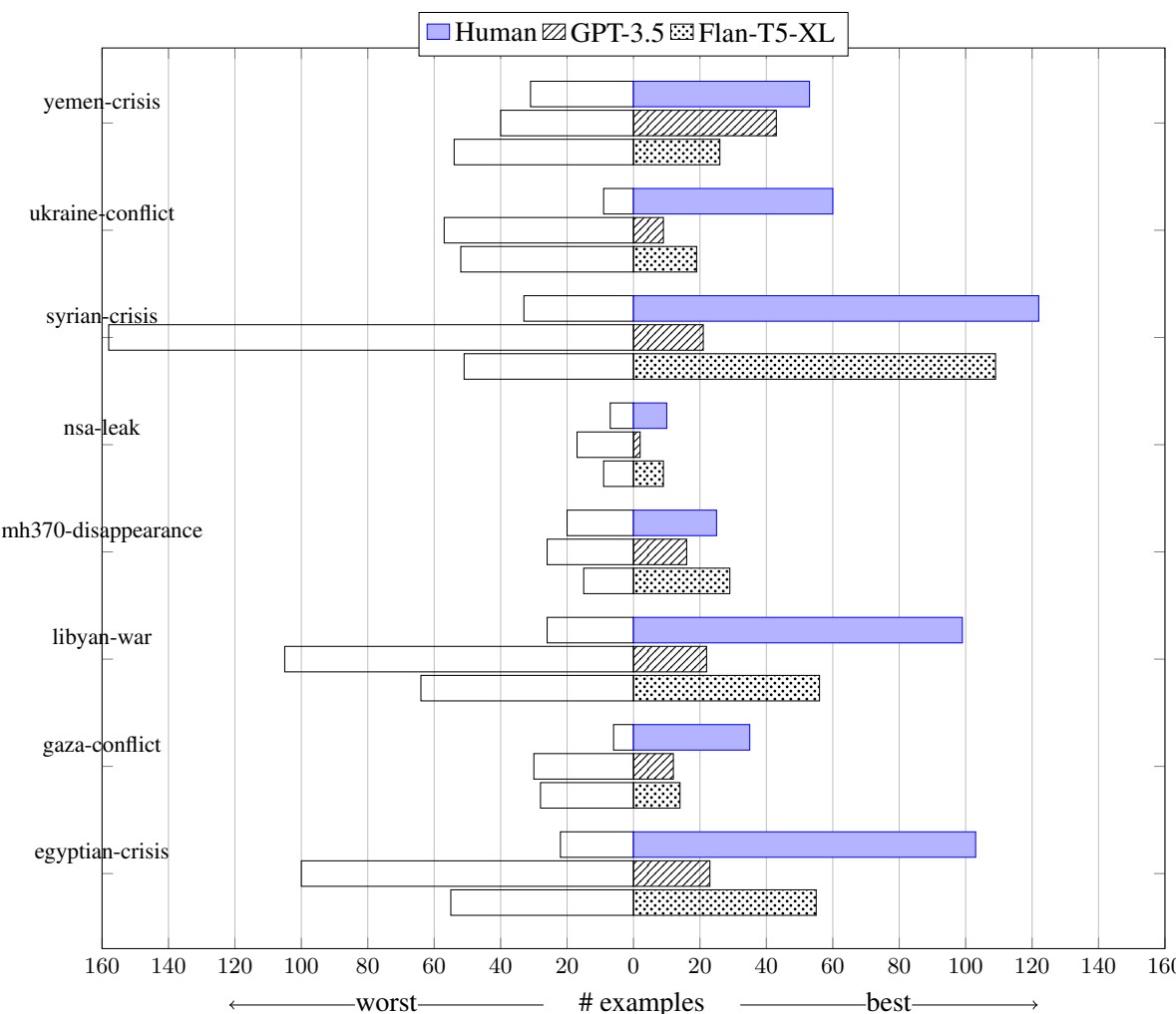

Figure 10: Aggregated best-worst votes for human-written, Flan-T5, GPT-3.5 backgrounds (BUS–GPT-4). The left and right halves report voted-worst and voted-best system counts respectively.

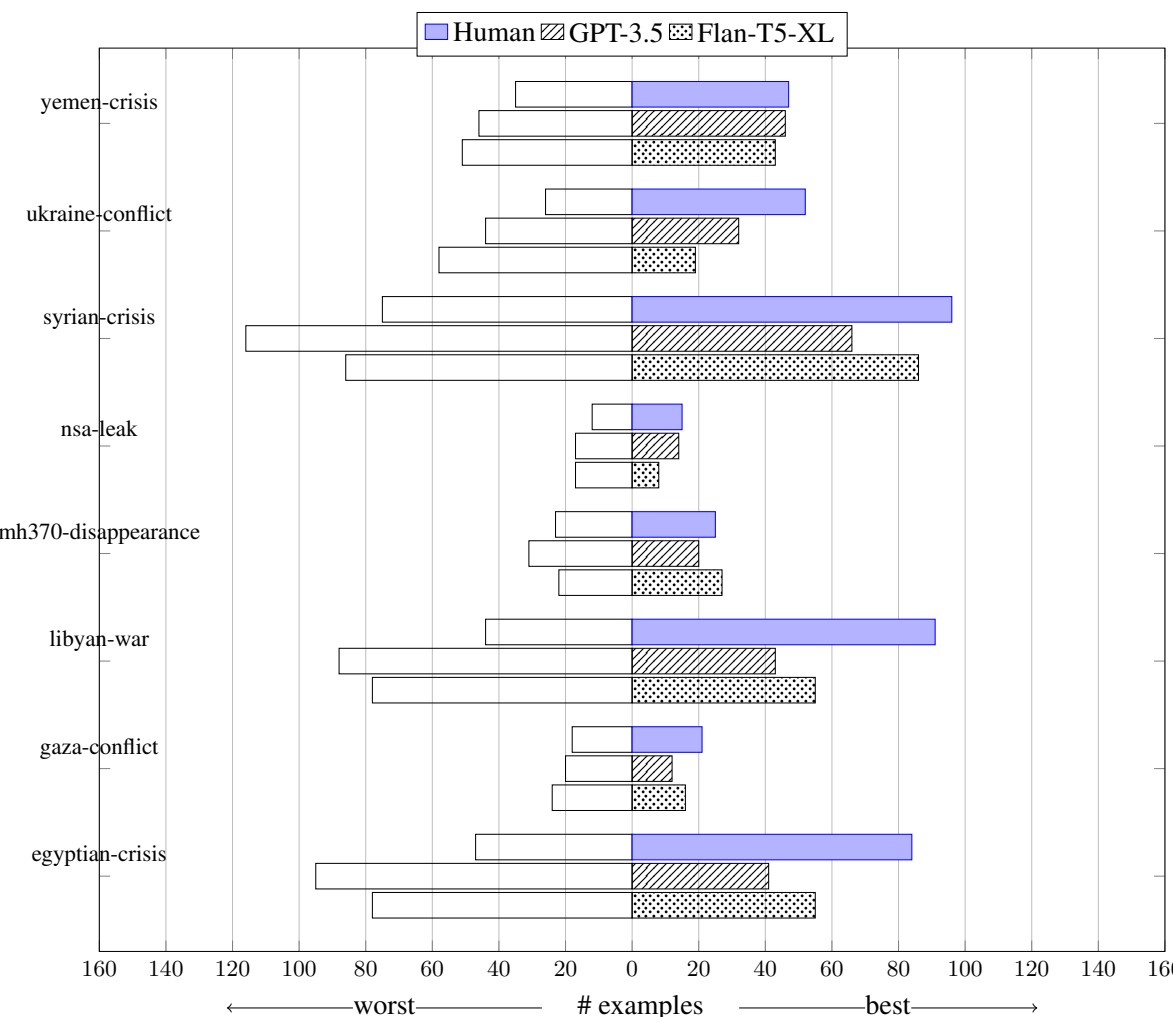

Figure 11: Aggregated best-worst votes for human-written, Flan-T5, GPT-3.5 backgrounds (BUS–Human). The left and right halves report voted-worst and voted-best system counts respectively.

*Date*: April 24, 2010.

*Update*: For the first time, oil is found to be leaking from the well. Pressure and release rate are unknown. A homeland security report on critical infrastructure says the problem has "no near-term impact to regional or national crude oil or natural gas supplies."

*Background*: (Human) On April 20th, BP-operated drilling rig Deepwater Horizon exploded in the Gulf of Mexico, 84km south-east of Venice, Louisiana, when a blowout preventer failed to activate, leaving 11 dead and 17 injured. The rig has been found upside down about a quarter-mile from the blowout preventer. Multiple federal departments and agencies are involved in 28 search-and-rescue operations covering about 5,300 square miles. Oil sheen has been reported, but the Coast Guard says it's had no indication that oil is leaking from the well 5,000 ft below the surface of the Gulf. nA homeland security department risk analysis has said the incident "poses a negligible risk to regional oil supply markets". White House press secretary Robert Gibbs played down any possible effects on President Obama's plans to expand offshore drilling, saying "I don't honestly think it opens up a whole new series of questions because, you know, in all honesty I doubt this is the first accident that has happened and I doubt it will be the last".

*Background*: (GPT-3.5) On April 20, 2010, the Deepwater Horizon drilling rig exploded in the Gulf of Mexico, leaving 11 workers missing and presumed dead, and injuring approximately 17. The blowout preventer failed to activate, and the rig sank to the bottom of the Gulf, potentially allowing oil to spew out from the well. The National Response Team, involving 16 federal departments and agencies, was activated, and chemical dispersants were readied to break up the oil slick. The US coast guard suspended the search for the missing workers and reported no indication of oil leaking from the well. The incident was deemed to pose a negligible risk to regional oil supply markets and was not expected to cause significant national economic impacts.

*Background*: (Flan-T5-XL) The Deepwater Horizon drilling rig exploded on April 20, 2010 while under construction in the Gulf of Mexico, killing 11 workers. The rig was found upside down about a quarter-mile from the blowout preventer. The force of the explosion broke off the rig's drillpipe, potentially allowing oil to spew out into the gulf. A National Response Team, involving 16 federal departments and agencies including the White House, Coast Guard, the departments of Defense, Homeland Security and Interior, and Environmental Protection Agency, among others, was activated. A Coast Guard official said the Macondo well, which the rrig had been drilling, could be releasing up to 8,000 barrels of oil per day. A remotely operated vehicle with camera went underwater and confirmed no visible oil flow from the well, but there were reports of an oil slick five miles long. Chemical dispersants were readied to break it up. A homeland security department risk analysis said the incident "poses a negligible risk to regional oil supply markets and will not cause significant national economic impacts". White House press secretary Robert Gibbs was asked whether the incident might affect Obama's plans to expand offshore drilling. He said, "I don't honestly think it opens up a whole new series of questions because, you know, in all honesty I doubt this is the first accident that has happened and I doubt it will be the last"

Table 8: This table provides the background summaries generated by humans, GPT-3.5, and Flan-T5 for the given update. See Figure 2 for the QA pairs generated from these update-background pairs by prompting `gpt-3.5-turbo`.

|  | ROUGE-1 | ROUGE-2 | ROUGE-L | QuestEval | BERTScore P | BUS–GPT-3.5 |
|---|---|---|---|---|---|---|
| Flan-T5-XL | 43.5 | 20.4 | 39.9 | 31.2 | 86.3 | 46.0 |
| Long-T5-TGlobal-XL | 40.1 | 16.7 | 36.4 | 33.4 | 86.9 | 46.2 |

Table 9: A comparison of Flan-T5 and Long-T5 systems on the dev set.

*Date*: June 4, 2014.

*Update*: Australian researchers release a recording of an underwater sound that could have been MH370 hitting the water.

*BUS–human* (Turker 1)
Q1: What is MH370?
Q2: What was the name of the researchers?
Q3: Where was the crash?
Q4: What else could the sound have been?
Q5: How did they record the sound?

*BUS–human* (Turker 2)
Q1: What was the flight path of MH370?
Q2: What were the last known communications from MH370?
Q3: What are the search parameters being used by the Australian researchers?
Q4: What are the other possible explanations for the underwater sound?
Q5: What are the implications of the underwater sound for the search for MH370?

*BUS–GPT-3.5*
Q1: What is MH370 and why is it significant?
Q2: How did the Australian researchers obtain the recording of the underwater sound?
Q3: What other evidence has been found regarding the disappearance of MH370?
Q4: What is the current status of the investigation into the disappearance of MH370?
Q5: What impact could this new evidence have on the families of the passengers and crew on board MH370?

*BUS–GPT-4*
Q1: What is MH370 and what happened to it?
Q2: Who are the Australian researchers involved in this investigation?
Q3: How were the researchers able to capture this underwater sound?
Q4: What evidence suggests that this sound could be MH370 hitting the water?
Q5: Has this new evidence brought any significant progress in the investigation of MH370?

Table 10: For an update from the 'MH370 flight disappearance' event, this table provides BUS questions generated by humans (MTurk), GPT–3.5 and GPT-4.

*Date*: November 23, 2011.

*Update*: President Saleh signs a deal brokered by the Gulf Cooperation Council which will see him hand over power to his vice president in exchange for immunity from prosecution for him and his family. He will remain honorary president for 90 days.

*BUS–human* (Turker 1)
Q1: What is Saleh president of?
Q2: What is the Gulf Cooperation Council?
Q3: What crimes is Saleh immune to prosecution for?
Q4: Who is Saleh's vice president?
Q5: What else was in the deal that was signed?

*BUS–human* (Turker 2)
Q1: What were the reasons for the protests against President Saleh?
Q2: What was the role of the Gulf Cooperation Council in mediating the deal?
Q3: What were the terms of the deal that President Saleh signed?
Q4: What was the reaction of the Yemeni people to the deal?
Q5: What are the long-term implications of the deal for Yemen?

*BUS–GPT-3.5*
Q1: What led to President Saleh's decision to hand over power to his vice president?
Q2: Who is President Saleh's vice president and what is his political background?
Q3: What is the Gulf Cooperation Council and how did they broker this deal?
Q4: What kind of immunity will President Saleh and his family receive and what crimes are they being protected from?
Q5: What will happen after the 90-day period when President Saleh's honorary presidency ends?

*BUS–GPT-4*
Q1: Who is President Saleh and what country is he the president of?
Q2: What led to the need for President Saleh to hand over power to his vice president?
Q3: What is the Gulf Cooperation Council and what role did they play in brokering this deal?
Q4: Why is immunity from prosecution a part of the deal for President Saleh and his family?
Q5: Who is the vice president that will be taking over power and what is their political background?

Table 11: For an update from the 'Yemen crisis' event, this table provides BUS questions generated by humans (MTurk), GPT–3.5 and GPT-4.

*Date*: November 22, 2011.
*Update*: Libya's interim prime minister Abdel Rahim al-Keeb announces a new cabinet.

*BUS–human* (Turker 1)
Q1: What led to the need for a new cabinet announcement in Libya?
Q2: Who was the previous prime minister ?
Q3: What challenges did the interim government face in forming the new cabinet?
Q4: What are the key responsibilities and goals of the new cabinet ?
Q5: How was the new interim prime minister selected?

*BUS–human* (Turker 2)
Q1: What happened to Libya's previous prime minister?
Q2: Can an interim prime minister create a new cabinet?
Q3: Do the people of Libya like Abdel Rahim al-Keeb?
Q4: When will there be a new permanent prime minister?
Q5: What happened to Libya's previous cabinet?

*BUS–GPT-3.5*
Q1: Who is Abdel Rahim al-Keeb and how did he become Libya's interim prime minister?
Q2: What were the reasons for the formation of a new cabinet in Libya?
Q3: Who are the members of the new cabinet and what are their qualifications and experience?
Q4: What are the main challenges facing the new cabinet in Libya and how do they plan to address them?
Q5: What is the current political situation in Libya and how does the new cabinet fit into the broader political landscape?

*BUS–GPT-4*
Q1: Who was the previous prime minister of Libya and why did they leave the position?
Q2: What led to Abdel Rahim al-Keeb becoming the interim prime minister?
Q3: What is the political situation in Libya that necessitated a new cabinet?
Q4: What are the main challenges facing the new cabinet in Libya?
Q5: Who are the members of the new cabinet and what are their political backgrounds?

Table 12: For an update from the 'Libyan crisis' event, this table provides BUS questions generated by humans (MTurk), GPT–3.5 and GPT-4.

*Date*: June 16, 2012.

*Update*: Escalating violence leads to the suspension of the UN observer mission in Syria.

*Background (GPT-3.5)*: The article provides a timeline of events during the Syrian uprising from May 2011 to May 2012. It includes the imposition of sanctions on Assad and senior members of the government by the EU, violent clashes and protests, defections, and the Arab League peace plan. The article also covers the UN-brokered cease-fire and the presence of UN observers, as well as the continued violence and massacres that occurred during this period.

---

*BUS–GPT-3.5*

Q1: What was the purpose of the UN observer mission in Syria?
Q2: What is the current level of violence in Syria and who is responsible for it?
Q3: How long has the UN observer mission been in Syria and what progress have they made?
Q4: What actions have been taken by the international community to address the violence in Syria?
Q5: What are the potential consequences of suspending the UN observer mission in Syria?

A1: The purpose of the UN observer mission in Syria was to monitor the implementation of a UN-brokered cease-fire.
A2: The article does not provide information on the current level of violence in Syria or who is responsible for it.
A3: The UN observer mission has been in Syria since April 2012, and the article does not provide information on the progress they have made.
A4: The international community has imposed sanctions on Assad and senior members of the government by the EU, and the Arab League has proposed a peace plan. The UN has also brokered a cease-fire and sent observers to monitor it.
A5: The potential consequences of suspending the UN observer mission in Syria are not discussed in the background text.

---

*BUS–GPT-4*

Q1: What is the UN observer mission in Syria and what is its purpose?
Q2: What specific events led to the escalation of violence in Syria?
Q3: How long has the UN observer mission been in Syria?
Q4: What are the potential consequences of suspending the UN observer mission in Syria?
Q5: Who are the main parties involved in the violence in Syria?

A1: Unanswerable
A2: Unanswerable
A3: Unanswerable
A4: Unanswerable
A5: Unanswerable

---

Table 13: For an update from the 'Syrian Crisis' event, this table provides BUS questions and answers extracted GPT–3.5 and GPT-4. BUS–GPT-3.5 suffers from answer hallucination, i.e., responds with an answer even if its not mentioned in the background text. See A1, A3 in BUS–GPT-3.5.