# OpenReview forum: "Background Summarization of Event Timelines"
_EMNLP/2023/Conference — EMNLP 2023 Main_

### Official Review · Reviewer_EUZX · 2023-08-02

**Soundness:** 4

**Excitement:**

3: Ambivalent: It has merits (e.g., it reports state-of-the-art results, the idea is nice), but there are key weaknesses (e.g., it describes incremental work), and it can significantly benefit from another round of revision. However, I won't object to accepting it if my co-reviewers champion it.

**Missing References:**

No missing references have been identified in the paper. However, it is noted that the authors skip citing relevant works related to timeline summarization, focusing only on datasets.

**Paper Topic And Main Contributions:**

The paper proposes a new task, namely Background summarization of event timelines. It introduces a dataset constructed by merging existing timeline datasets and enriched with manually collected information by human annotators. The paper employs standard metrics alongside a novel metric called BUS (Background Utility Score) for evaluation. The BUS metric assesses whether the background summary effectively helps answer reader questions.

The main contributions of the paper are as follows:

New Task Introduction: The paper introduces the task of background news summarization, recognizing the need to complement timeline updates with relevant background summaries for each timestep of a news event.

Data Collection: To facilitate research in this new task, the paper constructs a dataset by merging existing timeline datasets and enhancing it with background summaries written by human annotators for each timestep of each news event. This dataset provides a valuable resource for further exploration and development of background summarization systems.

Evaluation Metric: The authors propose a novel evaluation metric called Background Utility Score (BUS) to assess the quality of background summaries.

**Questions For The Authors:**

A: Regarding the BUS score, how do you plan to address the potential bias introduced by using automatically generated questions from GPT-3.5? Have you considered alternative approaches for generating evaluation questions to mitigate this bias?

B: Could you provide further justification for the choice of a maximum length of 3696 for GPT-3.5 during the evaluation? How does this selection impact the performance comparison across different models?

C: The choice of a split ratio of 3/3/8 for train/dev/test data is unusual in the context of natural language processing tasks where the test set is typically smaller than the training set. Could you please provide a justification for this specific split ratio and explain why the test set is significantly larger than the training set and development set?

**Reasons To Accept:**

1. The paper introduces a new and interesting task of background news summarization. This novel task has the potential to attract the interest of researchers in the field of NLP and temporal information analysis.

2. The paper creates a new data collection for the task. This resource can serve as a benchmark dataset for future research.

**Reasons To Reject:**

1. The potential bias introduced by the BUS score, as it relies on questions automatically generated by GPT-3.5. The bias in question generation could impact the overall evaluation and may not accurately reflect the true utility of the background summaries. The experiments show variations when human-generated questions are used, which further raises concerns about the reliability and generalizability of the metric (as also suggested by the authors).
[EDIT: Addressed by reviewers, results may be relevant and should be included in the revised version if possible]
2. The evaluation using different models with different maximum lengths, such as GPT-3.5 with a max length of 3696, may significantly influence the performance of the summarization systems. The choice of maximum length might not be sufficiently justified, and variations in model capabilities could obscure the fair comparison of different methods.
[EDIT: Addressed by reviewers]
3. Section 4, which discusses the use of Named Entity Recognition (NER) to generate questions, lacks clarity and needs improvement. The paper should provide a more comprehensive explanation of why NER is employed and how it contributes to the generation of relevant questions.
[EDIT: Addressed by reviewers, maybe a few lines of the response may be needed in the paper also.]

**Reproducibility:**

4: Could mostly reproduce the results, but there may be some variation because of sample variance or minor variations in their interpretation of the protocol or method.

**Reviewer Confidence:**

4: Quite sure. I tried to check the important points carefully. It's unlikely, though conceivable, that I missed something that should affect my ratings.

---

> ### Author Rebuttal · Authors · 2023-08-29
>
> We thank the reviewer for their thoughtful and constructive comments. We are glad the reviewer acknowledged the importance of our proposed background summarization task as well as our dataset contribution.
>
> > how do you plan to address the potential bias introduced by using automatically generated questions from GPT-3.5?
>
> As correctly pointed out by the reviewer, human-based BUS varies from GPT-3.5-based BUS. Human-based utility judgments are ideal and generally accepted as the gold-standard, but they are expensive to collect and are difficult to use during model development. GPT-3.5-based BUS is an efficient and cost-effective alternative that is useful for picking the best summarization systems during development and LLM-based evaluation has gained recent traction in the community (see below).
>
> New results: Specifically, to account for any bias of using GPT-3.5-based BUS for evaluating GPT-3.5-based background summaries, we conducted a new BUS evaluation using GPT-4. This follows recent work in the literature that uses GPT-4-based evaluation (lines 534–536 in our paper). We present the results below. We report best-worst scaling scores, i.e., the percentage of times a summary type is chosen as best minus the percentage of times it was chosen as worst (lines 451–460). Our results indicate that GPT-4-based BUS is more aligned with the BUS-human results as well as the human best-worst scores (BW) results:
>
> | Method | Human | Flan-T5-XL | GPT-3.5 |
> | :--- | :---: | :---: | :---: |
> | BW | 0.243 | -0.075 | -0.168 |
> | BUS--GPT-3.5 | 0.181 | -0.297 | 0.095 |
> | BUS--Human | 0.163 | -0.07 | -0.138 |
> | BUS--GPT-4 | 0.7 | -0.353 | -0.627 |
>
> We see the extensive human evaluations, both with human best-worst scores as well as the novel question-based BUS-Human evaluation method, as one of the contributions of our paper. We will point this out clearly in the next draft. These human evaluations led us to discover the bias inherent in GPT-3.5-based evaluations, which we will discuss in more detail in the paper as well. We believe that these findings will be a foundation for future work in automatically evaluating background summaries.
>
> We would also like to highlight that prior work on related tasks (Section 2) such as recaps of TV scripts and disentangled contribution-context of scientific papers did not include a utility-based metric. We hypothesize our metric can be applicable in these tasks where the recap should be relevant to the current episode and the context should be sufficiently related to the contributions of a paper.
>
> > Could you provide further justification for the choice of a maximum length of 3696 for GPT-3.5 during the evaluation?
>
> GPT-3.5 allows up to 4096 tokens for both the input and completion text. We allocate 400 tokens for the completion (background summary) and budget the remainder (3696 tokens) to the input/source text. We compare it against a long-context model (LongT5 [4096 tokens]) in our experiments. Our results indicate that GPT-3.5 outperforms LongT5 even though it uses slightly shorter input text.
>
> > The paper should provide a more comprehensive explanation of why NER is employed and how it contributes to the generation of relevant questions.
>
> We apologize for the lack of clarity about the use of NER systems in our query-focused setup. We consider two variants for queries. First, we use the full update text ($U_{t+1}$) as the query. Second, we first extract named entities from $U_{t+1}$ and use those keywords as the query. The named entity-based approach removes any potential noise from the update and focuses solely on extracting background information about important persons or locations specified in the update. As we show in Appendix A.6, both systems provide similar performance. We only included query ablations with Flan-T5 and not GPT-3.5 due to associated costs with running OpenAI API.
>
> > Could you please provide a justification for this specific split ratio and explain why the test set is significantly larger than the training set and development set?
>
> Thanks for raising this important question. Recent literature in text summarization showed strong abilities of few-shot and zero-shot summarization using large language models (Goyal et al., arXiv 2022, Zhang et al., arXiv 2023). Systems such as Flan-T5 and possibly GPT-3.5 are already instruction fine-tuned on standard summarization datasets such as CNN DailyMail, Multi News, etc., and therefore capable of performing summarization with limited or sometimes no additional training data.
>
> Considering these system capabilities, we decided to budget only a small fraction of expert-annotated data for training and development and leave most events to the test set for evaluation (Table 1). This allows sufficient data for further fine-tuning instruction models (Flan, GPT-3+) to our new task as well as maintaining sufficient diversity in the test set. All three splits include a mix of short and long timelines and the test set is mostly temporally separated from the train/dev events (Table 1).

---

### Official Review · Reviewer_o6pZ · 2023-08-03

**Soundness:** 4

**Excitement:**

3: Ambivalent: It has merits (e.g., it reports state-of-the-art results, the idea is nice), but there are key weaknesses (e.g., it describes incremental work), and it can significantly benefit from another round of revision. However, I won't object to accepting it if my co-reviewers champion it.

**Paper Topic And Main Contributions:**

The paper addresses the problem of event background summarization, which is defined as the task of giving all the necessary background to an event such that a reader can understand an update to that event. In particular the authors create a new dataset containing background summaries, they test background summarization generation on SOTA NLP models and finally present a metric to evaluate that the summary is useful for the task. I would say the major contribution is in dataset creation, with the note that the dataset is only in English.

**Questions For The Authors:**

A. A straightforward baseline for the background summary would be a more carefully constructed prompt for GPT3.5, which first asks the model to extract questions from the query update and then to answer the questions using the background, in the form of a paragraph. Basically your BUS metric. Have you tried this? Of course, this summary should then be evaluated by humans.

B. Regarding the BUS metric and the weakness that I emphasis above, can you please explain your reasoning in not enforcing at least some predefined questions.

C. How you verified if two different people came up with similar BUS questions?

D. ROUGE agreement is very low on the human generated backgrounds. While this is explainable,what is the guarantee that the task is not highly subjective and hence good performance cannot be assured?

**Reasons To Accept:**

I found the paper well written and easy to follow. In my opinion, the experimental evaluation rigorous. In addition, the problem studied is interesting and has at least one application in journalism.

**Reasons To Reject:**

The main weakness is the new metric BUS that the authors propose should be used to measure how good is a background summary for a given event update. The authors present two variants: GPT is used to propose questions based on the event update or a human is asked. Both system and humans do not receive any guidelines. While this gives freedom to deal with many types of text, it also means that the question-based metric can be very biased when done via GPT or via humans. Biased, in the sense that there are no formal requirements for at least some of the questions that the BUS metric should ask, hence asking two different people or rerunning GPT with slightly different parameters might give completely different questions. At least some questions could have been predefined if the event update did not answer them, and this are the standard questions that a summary should contain: (1) Who, (2) What, (3), When, (4) Where, and (5) Why. If the authors think enforcing these questions is inappropriate, they should argue this.

In addition, the same bias applies to the creation of the dataset, as the background summaries created by humans should contain the answer to these questions, when the answer is not present in the event update.

**Reproducibility:**

4: Could mostly reproduce the results, but there may be some variation because of sample variance or minor variations in their interpretation of the protocol or method.

**Reviewer Confidence:**

4: Quite sure. I tried to check the important points carefully. It's unlikely, though conceivable, that I missed something that should affect my ratings.

---

> ### Author Rebuttal · Authors · 2023-08-29
>
> We thank the reviewer for their thoughtful and constructive comments. We are glad the reviewer found our proposed task and experiments to be interesting and rigorous. Below, we address the reviewer’s concerns regarding one of our contributions, BUS.
>
> > The main weakness is the new metric BUS … Both system and humans do not receive any guidelines
>
> For the human-based BUS, we provide guidelines for both the question (Figure 6 in the appendix) and answer generation (Figure 7 in the appendix). Due to the space limitations, we could not include these figures in the main paper. For the GPT-based BUS, we provide task-level guidelines (see Table 6 in the appendix). Unlike the human-based BUS, we did not include examples of background questions in the GPT-based BUS. In our preliminary analysis, we found that the GPT-3.5-based system was capable of understanding the notion of a “background question” without any need for examples.
>
> > Biased, in the sense that there are no formal requirements for at least some of the questions that the BUS metric should ask … At least some questions could have been predefined if the event update did not answer them
>
> Questions about the background context are specific to the events described in an update. This could include a mix of question types as seen in the examples from Figures 2 and 6. Therefore, we do not expect constraints on the question types (why, what, how, etc.,) would necessarily lead to a better metric. However, we agree with the reviewer's concern about sampling different questions from GPT. This could induce some variance in the BUS scores. We believe an average score reported over five questions could be an approximate indicator of the background summary quality.
>
> > A straightforward baseline for the background summary would be a more carefully constructed prompt for GPT3.5 … Basically your BUS metric. Have you tried this?
>
> We have not tried this baseline yet, but it is an interesting approach. Since this needs additional human evaluation, we could not do this experiment in time for our response.
>
> > How you verified if two different people came up with similar BUS questions?
>
> Due to the limited evaluation budget, we could not collect questions from two or more annotators in human-based BUS. However, we acknowledge this is a valid point and related to the above point on variance in GPT-based BUS (pre-defined question types). We will try to collect expert or crowdsource questions for a small sample and report our findings in the next draft.
>
> > ROUGE agreement is very low on the human generated backgrounds.
>
> We acknowledge potential variance in background summaries for the same update. Therefore, we include three backgrounds for each update in our dataset. In our results, we also report average scores across the three backgrounds, minimizing the impact of subjectivity in the process.

---

### Official Review · Reviewer_Lm5f · 2023-08-03

**Soundness:** 4

**Excitement:**

4: Strong: This paper deepens the understanding of some phenomenon or lowers the barriers to an existing research direction.

**Paper Topic And Main Contributions:**

The paper proposes a new summarization task of creating a background summary from event timelines, which are a sequence of very brief texts describing key events for a major event that has been curated by journalists. To support this, a dataset was created by combining timeline datasets from three existing datasets and then labeled using Mechanical Turk workers pre-screened for reliability. To provide benchmarks for the task, experiments comparing three LLM summarization systems for generating both generic summaries and query-focused summaries found that generic summaries were best. Also to support the task, a query-based evaluation, Background Utility Score (BUS) was proposed. Human evaluation comparing the different query-focused LLM-generated summaries with human-created background summaries was performed; the evaluation results found the LLM-generated questions used as prompts for summaries did not produce results resembling those from human-generated questions, motivating the use of Mechanical Turk workers for generating questions instead of an LLM, which was tested and observed to perform better.

**Questions For The Authors:**

Related to utility of the task, in line 240, it is stated that news articles “end with necessary background information”. Couldn’t the background information from a recent article be used as the background summary, perhaps together with a brief summary from the current article? Have you compared a summary using the article’s background summary to the timeline summaries?

**Reasons To Accept:**

Proposed the task of background summarization.

Merged several timeline datasets and had humans annotate them with a background summary for each time step to create a new dataset that will be made available.

Proposed a query-based evaluation task

Benchmark evaluation showed that the task is difficult—using state-of-the-art summarizers GPT-3.5 and Flan-T5-XL both performed worse in generating a query-focused summary than in generating a generic summary to serve as a background summary.

Evaluation showed that humans generated better queries for evaluating using the proposed BUS evaluation measure than using GPT-3.5.

The paper’s strength is the setup of a new summarization task which includes a dataset, code, an evaluation method, and benchmark performance. The benchmark evaluation shows that the task is difficult.

**Reasons To Reject:**

The paper mentions that journalists create timelines for major events. The paper would be strengthened if more applications of the task could be given, to indicate more general utility than just journalists.

For the evaluations comparing summaries, only generated summaries were compared in Figs 3 and 4. I would like to see generic summaries (a concatenation of the updates that performed better than the background summary in Table 3) compared to the generated summaries in the figures.

**Reproducibility:**

5: Could easily reproduce the results.

**Reviewer Confidence:**

4: Quite sure. I tried to check the important points carefully. It's unlikely, though conceivable, that I missed something that should affect my ratings.

**Typos Grammar Style And Presentation Improvements:**

The difference between recaps and background summary needs more differentiation (lines 185-213).  They seem similar, except in recaps the data is episodes and in background summary, the data is updates:

- Recaps typically present a summary of the past episode(s) that are relevant to events detailed in the current episode.
and
- Our definition of background constitutes a summary of previous updates in a given event that are directly relevant to the current, most recent update.

Would you like to emphasize the query aspect as a differentiator?

---

> ### Author Rebuttal · Authors · 2023-08-29
>
> We thank the reviewer for their thoughtful and constructive comments.
>
> > The paper would be strengthened if more applications of the task could be given, to indicate more general utility than just journalists.
>
> In addition to complementing journalists who create timelines, a background summarization system can assist readers in following complex news events by providing necessary contextual information. An interesting application would be to contextualize short-text content such as tweets with background information extracted from news articles about the events. In addition to providing much-needed context, this can be useful for verifying the factuality of the events described in the tweet itself (e.g., Twitter community notes). Finally, in conversational AI settings, a background summary can inform a “tell me what I need to know to understand this event” request in news-centric experiences.
>
> > For the evaluations comparing summaries, only generated summaries were compared in Figs 3 and 4.
>
> In our evaluation setup (Figures 3 and 4), we compare the Flan-T5 and GPT-3.5 models from the generic setup (top half of Table 3) and the human-annotated backgrounds (section 3.2). We chose to evaluate the generic systems instead of query-focused systems due to their superior performance on ROUGE and factuality metrics on the development set. We will add a clarification in lines 433–435 to explicitly state this.
>
> > Couldn’t the background information from a recent article be used as the background summary
>
> In general, we agree with the idea of using background content directly from the news articles to construct a background for the entire event. In this work, we focused on extending the existing work on timeline summarization, and therefore, we reused the information already present in previous update summaries. However, we expect future work would focus on including the news articles and content therein to construct the background without the need for timelines.
>
> > The difference between recaps and background summary needs more differentiation (lines 185-213)
>
> As accurately pointed out by the reviewer, episode recaps and news backgrounds are closely related. A key distinction is that the recap typically only provides information from the most recent episode(s), and doesn’t provide general context to the story. In contrast, background often provides a quick overview of the event that could include information from the very first update.

---

### Meta-Review · Area_Chair_guML · 2023-09-11

**Recommendation:** 4

**Metareview:**

This paper introduces the task of background summarization from event timelines and proposes an experimental framework that includes a benchmark dataset and a dedicated evaluation metric. Overall, reviewers are all positive about this work after the authors's responses and discussion. They agree on the the contribution and novelty of the paper (interesting new task, introduced human-annotated dataset). Reviewers raised a concern about the potential bias/weakness of the introduced evaluation metric that was discussed in the rebuttal.

---

### Decision · Program_Chairs · 2023-10-07

**Decision:**

Accept-Main

**Comment:**

This paper introduces the task of background summarization from event timelines and proposes an experimental framework that includes a benchmark dataset and a dedicated evaluation metric. Overall, reviewers are all positive about this work after the authors's responses and discussion. They agree on the the contribution and novelty of the paper (interesting new task, introduced human-annotated dataset). Reviewers raised a concern about the potential bias/weakness of the introduced evaluation metric that was discussed in the rebuttal.